# Fermi surface tomography

Sergey Borisenko [1,2✉], Alexander Fedorov [1,2], Andrii Kuibarov[1], Marco Bianchi[3], Volodymyr Bezguba [1,4], Paulina Majchrzak [3], Philip Hofmann[3], Peter Baumgärtel [5], Vladimir Voroshnin[5], Yevhen Kushnirenko[1], Jaime Sánchez-Barriga [5,6], Andrei Varykhalov [5], Ruslan Ovsyannikov[5], Igor Morozov[1], Saicharan Aswartham [1], Oleh Feia[1,4], Luminita Harnagea [1], Sabine Wurmehl[1], Alexander Kordyuk[4], Alexander Yaresko [7], Helmuth Berger[8] & Bernd Büchner[1,9]

Fermi surfaces are essential for predicting, characterizing and controlling the properties of crystalline metals and semiconductors. Angle-resolved photoemission spectroscopy (ARPES) is the only technique directly probing the Fermi surface by measuring the Fermi momenta ($k_F$) from energy- and angular distribution of photoelectrons dislodged by monochromatic light. Existing apparatus is able to determine a number of $k_F$-vectors simultaneously, but direct high-resolution 3D Fermi surface mapping remains problematic. As a result, no such datasets exist, strongly limiting our knowledge about the Fermi surfaces. Here we show that using a simpler instrumentation it is possible to perform 3D-mapping within a very short time interval and with very high resolution. We present the first detailed experimental 3D Fermi surface as well as other experimental results featuring advantages of our technique. In combination with various light sources our methodology and instrumentation offer new opportunities for high-resolution ARPES in the physical and life sciences.

[1] Leibniz Institute for Solid State and Materials Research, IFW Dresden, D-01171 Dresden, Germany. [2] Fermiologics, D-01069 Dresden, Germany. [3] Department of Physics and Astronomy, Interdisciplinary Nanoscience Center (iNANO), Aarhus University, 8000 Aarhus C, Denmark. [4] Kyiv Academic University, 03142 Kyiv, Ukraine. [5] Helmholtz-Zentrum Berlin für Materialien und Energie, BESSY II, 12489 Berlin, Germany. [6] IMDEA Nanoscience, 28049 Madrid, Spain. [7] Max Planck Institute for Solid State Research, 70569 Stuttgart, Germany. [8] Institute of Condensed Matter Physics, Ecole Polytechnique Fédérale de Lausanne, Lausanne, Switzerland. [9] Institute for Solid State and Materials Physics, TU Dresden, 01062 Dresden, Germany. ✉email: S.Borisenko@ifw-dresden.de

One hundred years ago, an experimental possibility to detect electrons emitted at a certain angle led to the development of one of the most revealing tools in condensed matter physics—angle-resolved photoemission spectroscopy (ARPES)[1]. ARPES has led to many important discoveries, most recently topological insulators[2], and has become a leading tool for the study of quantum materials[3]. The technique itself is relatively simple—one usually counts the outgoing electrons as a function of their kinetic energy, direction and photon energy—four variables in all. However, the experimental equipment remains quite demanding: variable photon energies require sophisticated light sources including synchrotron radiation, while atomically clean surfaces require an ultra-high vacuum, which severely limits the detection equipment. As a consequence, the cost of a relatively ordinary ARPES setup easily exceeds six figures. Currently, acquiring a single high-resolution 2D cut through the most important and succinct feature of the electronic structure—the Fermi surface (FS)—takes up a significant portion of the typically allocated synchrotron beamtime thus leaving 3D Fermi surfaces of the materials only partially explored. This, in particular, restricts a detailed comparison with the widely available nowadays calculated 3D Fermi surfaces.

In this work we demonstrate that using the novel approach, based on the Fourier electron optics combined with the retardation field of the detector, it is possible to detect photoelectrons close to the Fermi level in a more simple and efficient way. We apply the technique to study two superconductors, topological insulator and several other materials and find new important details of their electronic structure.

## Results and discussion

In Fig. 1a we show the 3D FS obtained with our novel technique. One hundred high-resolution 2D FS maps were taken within one hundred minutes using the synchrotron light with variable photon energies. The experimental data are compared with those calculated in the framework of density functional theory FS (Fig. 1b). Despite the general qualitative agreement, the differences in the size and curvatures of the individual layers emphasize the need to perform such measurements. It is the exact 3D shape of the Fermi surface that defines physical properties of the material, such as electrical or optical conductivities, elasticity, thermal expansion, magnetic susceptibility, etc. The data set underlying the 3D FS shown in Fig. 1a consists of more than 17 million data points, allowing the corresponding intensity distribution to be viewed in any two- or one-dimensional section of this volume in k-space, revealing even smallest details of the 3D FS.

Why was it possible to record such a data set? The aim of the ARPES experiment is to obtain the intensity distribution of the photoelectrons flying out of the surface in different directions and with different kinetic energies with the best possible angular and energy resolution. In general, applying an electric or magnetic field to the photoelectrons would reduce the resolution because the elements inducing these fields are never geometrically perfect. To keep the experimental setup minimal and assume a point source of electrons, the goal can be achieved by simply placing a hypothetical detector, sensitive to both position and energy, in front of the sample under investigation. The first reality is that the photon beam always leads to a finite region on the sample that emits the photoelectrons. Even with such an advanced detector, the angular resolution is limited by the size of this area and the distance to the sample, as the same spot on the detector can now be reached by electrons with different emission angles.

We overcome this problem by placing an electron lens between the sample and the detector (Fig. 2). Such a lens, also known as *einzel lens*, works similarly to an optical lens, which is a textbook example of a Fourier transform engine. In other words, all electrons leaving the sample at the same angle can be focused on a single point in the back focal plane of the lens. In Fig. 2 we show schematically how the focusing works for two angles: normal emission (green rays) and 10° (blue rays). As a position-sensitive detector, we use a micro-channel plate (MCP) consisting of millions of channels, each of which is capable of detecting the incoming electrons. Although in this arrangement the angular distribution of the photoelectrons can be detected, each channel of the MCP would detect all electrons flying in the same direction, regardless of their kinetic energy, i.e. so far we only have position- but not energy-sensitive detection system.

There are many ways to sort electrons according to their kinetic energy. Again, we try to keep the design minimal and use probably the simplest method. We apply a negative potential to the front surface of the MCP itself, which repels all electrons that have a lower energy than that set by this retarding potential (red beams in Fig. 2). This method is particularly advantageous when measuring the distributions of electrons from the Fermi level - precisely because there are no photoelectrons with the kinetic energy greater than that defined by Einstein's equation for the

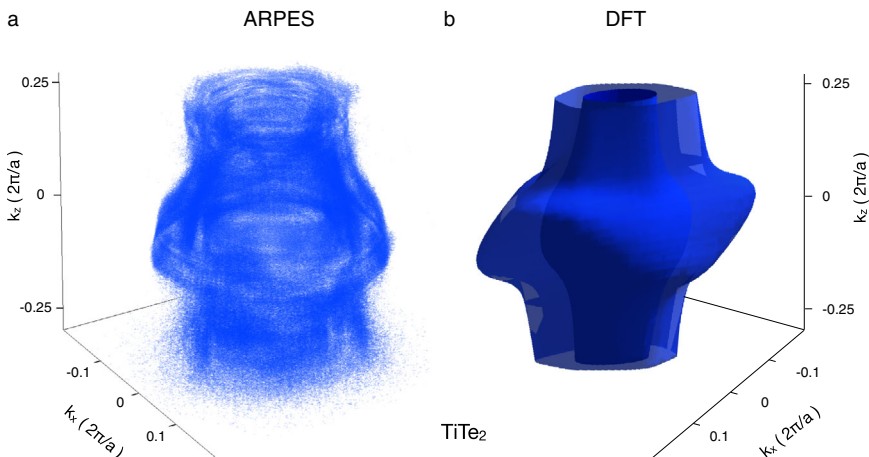

**a** ARPES **b** DFT

$k_x$ ($2\pi/a$) $k_z$ ($2\pi/a$) TiTe$_2$

**Fig. 1 Three-dimensional Fermi surface of TiTe$_2$. a** A voxelgram of photoemission intensity recorded using FeSuMa analyzer and synchrotron light. Photon energies between 14 and 34 eV are scanned with 0.2 eV step. Voxels within normalized intensity interval between 0.012 and 0.020 are shown. Total number of points of the underlying dataset is 1.71924e + 07. Average intensity is 0.007, absolute maximum is 0.041. **b** Calculated Fermi surface of TiTe$_2$ within the same momentum volume.

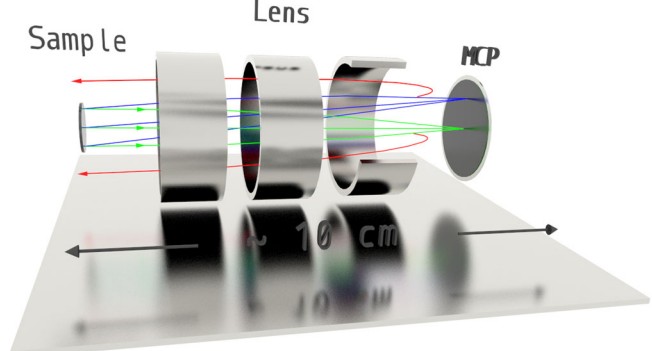

**Fig. 2 Simple ARPES.** Schematics of the method is shown. Electrons originate at the sample surface. Green rays correspond to normal emission. Blue rays represent electrons emitted at 10° angle in the vertical plane. Red rays are the electrons having lower kinetic energies from both beams, deflected by the retarding field. Three cylinders represent the electron lens. MCP is the position-sensitive detector.

photoelectric effect. By setting the threshold energy, say a few meV lower than this cutoff, we are able to obtain the desired angular distribution of photoelectrons from the Fermi level. It is important to mention that the retarding field of the MCP obviously modifies the focusing field of the lens and a special solution should be found for the potential pattern of all elements involved to get the desired result. An interactive animation (https://webdemo.3dit.de/ifw/detektor/) allows a better understanding of our concept.

In the following, we show some examples of the angular maps taken with the FeSuMa (Fermi Surface Mapper) spectrometer, designed and constructed as explained above[4], in Fig. 3. Due to the almost instantaneous result—one can see the FS contour live on the monitor (Supplementary Movie)—our technique offers numerous possibilities to vary individual experimental parameters while keeping all others constant. In Fig. 3a we start with presenting the photon energy dependence in TiTe$_2$. This is perhaps the most important ARPES dataset, out of all presented, as it provides a direct window into 3D k-space. We have tested FeSuMa at two synchrotron light sources in the broad photon energy range between 14 and 105 eV and always observed high-resolution angular distributions. The maps taken at lower photon energies shown in the left panels are similar to those underlying the 3D FS from Fig. 1a. Higher photon energy maps in Fig. 3a indicate the presence of further FS sheets that adopt the trigonal symmetry of the system. To fully capture these pockets, one can rotate the sample, as its angular position exceeds the lens acceptance of FeSuMa.

Figure 3b presents the results of a typical polarization-dependent experiment in topological insulator Bi$_2$Te$_3$. In this case, only the polarization of the incident laser light was switched from right-circular to left-circular. To emphasize the difference in the maps we present also the dichroism—the difference between the data taken using the light of opposite helicities. A classical picture caused by the spin-momentum locking of the topological surface states is observed (inset to Fig. 3, left panel). The duration of the whole experiment was less than a minute. The off-center position and elliptical shape of the contours are due to the finite tilt angle of the sample which could not be compensated by the manipulator with its limited rotational degrees of freedom. Taking this deviation into account, the map can be recalculated on the momentum scale (Supplementary Fig. 1).

Recently, it was demonstrated that iron-based superconductor LiFeAs ($T_c$ = 18 K) shows signatures of the nematic order upon entering the superconducting state[5]. One of them is the

deformation of the Fermi surface. The data from Fig. 3c clearly show that also the smallest FS sheet, which just fits into the angular window of FeSuMa when using the laser radiation, is deformed when the temperature changes from 20 K to 3 K, i.e. when the sample enters the superconducting state. The variations in intensity are best seen after subtracting the maps (inset to Fig. 3, right panel).

Two other parameters, the elapsed time after the cleavage of the single crystal and the position of the photon beam on the sample surface, are often crucial for understanding the ARPES spectra (Fig. 3d, e). Also in this case, the maps can be recorded successively while keeping all other conditions unchanged. This particular example shows the temporal evolution of the spectra in Bi$_2$Te$_3$ as well as a significant dependence of the signal on the probed region. Finally, the bottom row of panels (Fig. 3f) concludes the presentation of angular distributions showing the FS maps of various single crystals, ranging from high-temperature superconductors to topological insulators.

As mentioned earlier, there are four basic variables of ARPES: the kinetic energy, the photon energy and two angles that define the direction (we neglect here electron spin since its detection requires additional sophisticated apparatus). So far, we have limited ourselves to the electrons with the largest energies in the material - those near the Fermi level. The vast majority of phenomena in condensed matter physics depend crucially on the underlying dispersion of the electronic states, i.e. on how momentum distribution changes as a function of the binding energy. With our setup, this information is also available. This is the third new aspect of our technique and it is as simple as the previous two. The concept is illustrated in Fig. 4. We record a series of 2D angular distributions, each corresponding to the threshold energies lower than the one used to record FS map, thus scanning the energy interval of interest. As a result, we obtain a three-dimensional distribution of photoemission intensity, where each layer corresponds to the integrated intensity due to all electrons having kinetic energies more than the threshold one. The difference between two adjacent measurements (Supplementary Fig. 2) would represent the angular distribution corresponding to the particular binding energy. Thus, by differentiating such a data set along the energy direction, one can also determine the electronic structure below the Fermi surface (Supplementary Fig. 3).

Some of the most representative 2D momentum-energy slices from such 3D differential scans are shown in Fig. 5. Panels a correspond to high-symmetry directions of Bi$_2$Te$_3$ while panels b display the intensity distributions corresponding to the horizontal cuts close to the center of the angular map from Fig. 3b. The former clearly show the presence of the quantum well states in Bi$_2$Te$_3$ not only in the previously observed region of binding energy near the Dirac point (~400 meV), but also near the Fermi level, which explains the presence of multiple concentric FS from Fig. 3d.

Figure 5c, d represent the vertical cuts from Fig. 3c, f for BSCCO and LiFeAs samples respectively. We have clearly observed the opening of the anisotropic d-wave superconducting gap along the FS portions seen in Fig. 3f. One is able to track changes in the spectra caused by the lowering of the temperature either directly from the energy distribution curves (EDC) in Fig. 5f or as a shift of the leading edge midpoints in Fig. 5i. Even in the superconductor with much lower critical temperature of 18 K, LiFeAs, we were able to detect the corresponding shift of the leading edge (Fig. 5j) which indicated the opening of the superconducting gap of the order of 3 meV.

These observations are not surprising in view of the achieved overall energy and momentum resolutions. As the EDC shown in Fig. 5g demonstrates, the overall energy resolution achieved

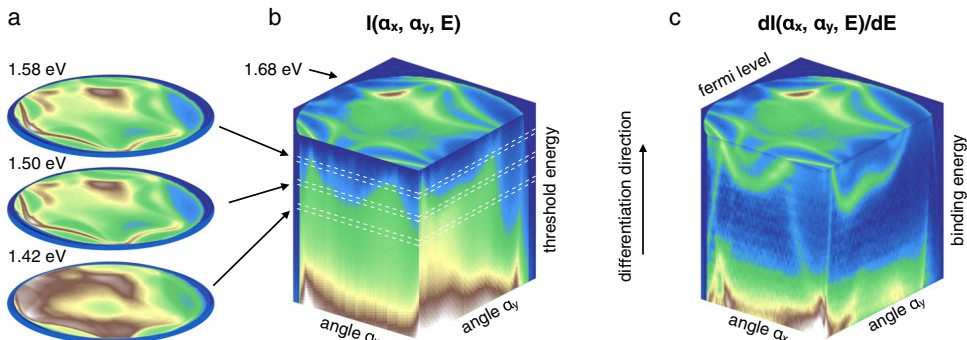

**Fig. 3 Two-dimensional Fermi surface maps.** Angular distributions, typically 500×500 pixels, of the photoemission intensity within the solid angle of ~30° and kinetic energy interval 5-20 meV in the vicinity of the Fermi level. **a** Angular distributions from $TiTe_2$ recorded at two synchrotron light sources. Lower energies - BESSY (6 K), higher energies - ASTRID2 (35 K). **b** Off-normal (~5°) emission laser (6 eV) data from $Bi_2Te_3$. recorded using left- and right-hand circularly polarized light, as well as their difference. $T = 3$ K. **c** Temperature-dependent data from LiFeAs and their difference. **d** Off-normal (~3°) emission data from $Bi_2Te_3$ recorded at different time after cleavage. $T = 3$ K. **e** The same as **d**, but taken earlier (after ~5 h) and from different positions at the cleaved surface. **f** Angular distributions from different materials. First four maps are recorded at BESSY at approximately 6 K. Two other maps—in the laboratory using the laser at 3 K.

**Fig. 4 Electronic dispersion from differentiation. a** Exemplary angular distributions of the intensity in $Bi_2Te_3$ recorded at different threshold energies. Angular map taken at 1.68 eV corresponds to the Fermi level. **b** Three-dimensional intensity distribution consisting of 73 layers measured with 10 meV steps. Background intensity above the Fermi level is not shown. Acquisition time ~30 min. Laser, 6 eV. T = 3 K. **c** Derivative of the dataset shown in **b** along energy axis. Top maps in **b** and **c** are the same since there is no intensity above the Fermi level.

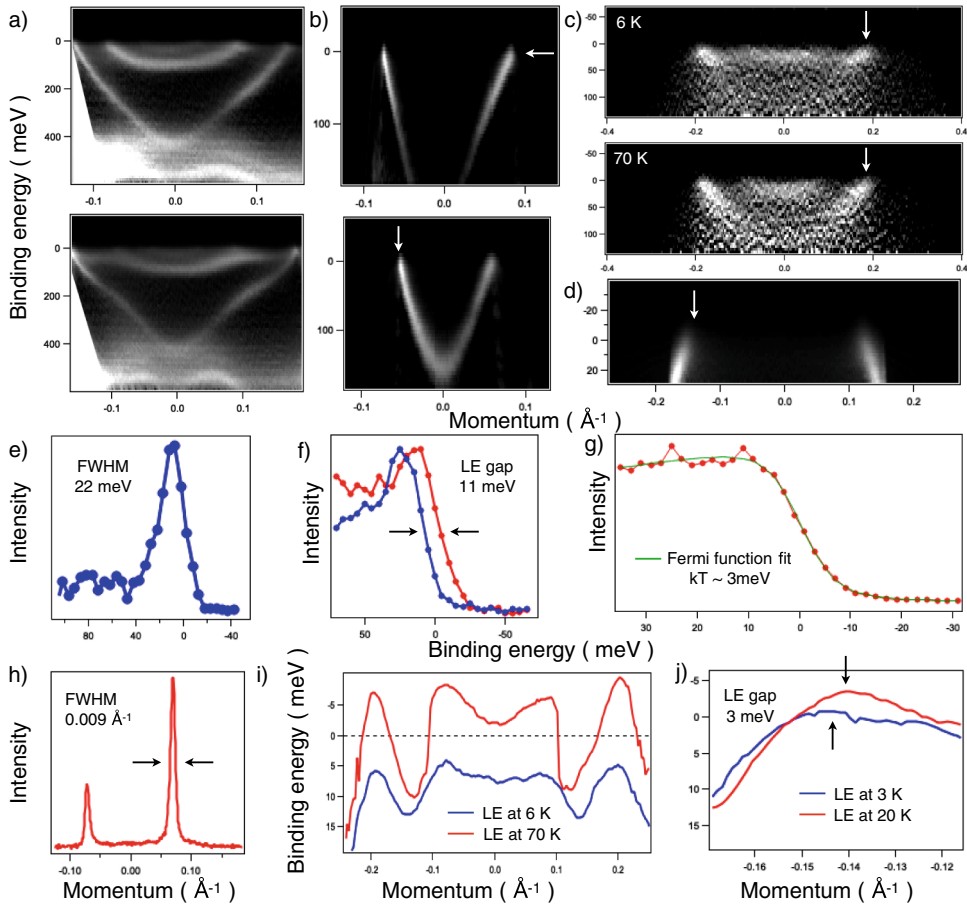

**Fig. 5 Energy direction and resolution. a** Momentum-energy cuts along the high-symmetry ΓK- and ΓM- directions of $Bi_2Te_3$ extracted from the differential scans shown in Fig. 4 and converted to the momentum scale. Laser, 6 eV. T = 3 K. Energy step−10 meV. **b** Same as **a**, but taken earlier after cleavage and with energy step of 5 meV. **c** Dispersion along the cut through the antinodes in BSCCO above and below superconducting transition ($T_c$ ~ 65 K). Synchrotron, 24 eV. Energy step−5 meV. Acquisition time of the whole 3D dataset ~20 min. Noise is due to variations of the ring current, which were not recorded and considered. **d** Vertical cut through the smallest FS of LiFeAs (Fig. 3c, 20 K). Energy step−2 meV. **e** EDC corresponding to white arrow in **b**. **f** EDCs corresponding to white arrows in **c** and integrated within ~0.05 Å$^{-1}$ window. **g** Background EDC from the momentum location shown by white arrow in **d** shortly after $k_F$ so that the spectrum looks like a Fermi edge. It is integrated within 10×10 pixels of the camera from the 500×500 dataset. There are 34 layers in this particular 3D wave—the energy steps taken every 2 meV. Each layer took 20 seconds to record. T = 3 K. **h** MDC from **b**. **i** Leading edge midpoints of the EDCs from **c**. **j** Leading edge midpoints from some of the EDCs in **d**.

so far is of the order of 12 meV. We believe this result can still be improved. Since we do not accelerate the photoelectrons in the lens and only minimally change their initial trajectories, the theoretical resolution exceeds that of conventional analyzers. The full width at half-maximum of the momentum distribution curve in Fig. 5h suggests that features in the momentum space can be resolved with better than a hundredth of a Brillouin zone precision.

The presented data sets imply that FeSuMa is a promising ARPES tool despite its simple design. Compared to existing hemispherical, time-of-flight and grating/display analyzers[6–8], it has a number of advantages. The absence of the entrance slit not only allows the angular distribution to be recorded in 2D (and thus much faster), but also guarantees isotropic angular resolution. The latter is anisotropic in hemispherical analyzers - the sampling of angles perpendicular to the slit is defined by the slit size itself, within which the signal must be integrated. This often leads to "stripy" FS maps (Supplementary Fig. 4), the recording of which requires further measurements involving either rotation of the sample or deflection of the electron beam within the lens by additional electron optical elements or by movement of the lens itself[9,10]. Such methods distort the original 2D angular distribution so that only a narrow strip of it can be recorded in each case.

Since the hemispheres cannot be placed in the immediate vicinity of the sample, the entrance slit is usually situated in the "second" focal plane of the lens, which inevitably causes the complexity of the lens itself.

Time-of-flight technique needs a pulsed light source, which results in longer measuring times and considerable space-charge effects, as well as sophisticated delay-line detectors, significantly complicating the operation of the instrument. Finally, the display analyzers typically involve a number of grids acting as energy filters, which results in a significantly worse momentum and energy resolution. Absence of the detailed and high-resolution experimental 3D Fermi surfaces in the literature (There are numerous ARPES studies, including ours, in which the 2D slices through the 3D Fermi surface (also along $k_z$) are shown, as well as various types of interpolations when the probing mesh is scarce in momentum space (at least an order of magnitude less detailed than in Fig. 1a). It is not possible to mention them all here) can be considered as an indicator of limited possibilities offered by the state-of-the-art ARPES equipment.

Among the further advantages of FeSuMa are the easiness of adjustment and operation. The open MCP allows to detect also the photons, which are scattered from the sample edges or imperfections, thus guaranteeing the detection of electrons from

the mirror-like portion of the studied surface—a requisite for high quality ARPES experiment. Such portion of the surface specularly reflects the photons and they do not enter the analyzer. The analyzer can work in a direct spatially-resolving mode imaging the source of electrons in two dimensions. This makes positioning of the source directly into the focus of the analyzer a matter of several seconds. Finally, a very compact and simple design—as is sketched in Fig. 2, detection of the photoelectrons can occur ~10 cm away from the source, - makes it possible to carry out high-resolution ARPES experiments at very low temperatures and at a fraction of usual costs. FeSuMa without cables would easily fit on this page.

Our technique also has fundamental implications. A few decades ago, angle-multiplexing analyzers enabled the transition from 1D EDCs to 2D momentum-energy maps, turning photoemission from a mere band-mapping tool into a sophisticated many-body spectroscopy. We hope that the addition of another dimension to ARPES will make the 3D spectral function accessible and reveal previously unknown aspects of electronic interactions and correlations. We believe that the increased efficiency of our instrument will provide the basis for high-throughput measurements necessary to match global computational efforts for material design, enable a new level of quality control of thin films and single crystals by complementing the usual crystal structure analysis with electronic structure analysis, and enable measurements of systems sensitive to radiation damage, such as organic crystals. Finally, the rapidly developing time-resolved and nano-derivatives of ARPES, which are beginning to successfully probe the electronic structure of quantum materials subjected to gate fields, currents, screening and strain in operando[11], should certainly benefit from the high informativity of Fermi surface tomography.

## Methods

We used single crystals of $TiTe_2$, $BiTeI$, $Bi_2Te_3$, $ZrTe_3$ and $(Pb,Bi)_2Sr_2CaCu_2O_{8+\delta}$ grown by Helmuth Berger. Growth procedure for each of them can be found elsewhere. LiFeAs single crystals were grown by self-flux using the standard method[12].

Self-consistent band structure calculations of $TiTe_2$ were performed using the linear muffin-tin orbital method in the atomic sphere approximation as implemented in PY LMTO computer code.

ARPES data have been collected using FeSuMa electron analyzer and synchrotron radiation from BESSY and ASTRID2 storage rings at beamlines U125-2_NIM (14–34 eV) and AU-SGM3 (20–105 eV) as well as laboratory-based 6 eV laser source.

## Data availability

The datasets generated and/or analyzed during the current study are available from the corresponding author on reasonable request.

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

## Acknowledgements

SVB is grateful to BMWi for the support within the WIPANO project. The work is supported by BMBF via UKRATOP program. AlK was supported by the National Research Foundation of Ukraine (Project 2020.02/0408). J.S.-B. acknowledges financial support from the Impuls- und Vernetzungsfonds der Helmholtz-Gemeinschaft under grant No. HRSF-0067. We thank HZB and ISA for the allocation of synchrotron radiation beamtimes at BESSY and ASTRID2. We thank Davide Curcio for the suggestion to implement the direct (spatial) mode in FeSuMa, Uwe Siegel, who successfully guided us through the technology transfer process, Ulrike Nitzsche for IT support, as well as Roland Hübel, Falk Sander and Frauke Thunig for invaluable technical support.

## Author contributions

S.V.B. invented the method and wrote the paper with contributions from all authors. S.V.B. and A.F. designed and constructed FeSuMa. S.V.B., A.F., An.K., M.B., V.B., P.M., P.H., Y.K., A.l.K. and B.B. proposed, planned and performed ARPES experiments. S.V.B., A.F., An.K. and P.M. analyzed the data. M.B., P.B., V.V., J.S.B., A.V. and R.O. provided support at the beamlines of the synchrotron light sources. I.M., S.A., L.H., S.W., H.B. and B.B. proposed, planned and performed synthesis of the single crystals. O.F. and A.Y. carried out band structure calculations.

## Funding

## Competing interests

The authors declare no competing interests.
