## [Peer Review File · Nature Communications]

Title: Fermi surface tomographyREVIEWER COMMENTS

Reviewer #1 (Remarks to the Author):

This manuscript presents a simple ARPES equipment mostly focusing on the Fermi surface mapping by photoemission; the authors name it FeSuMa (Fermi Surface Mapper). With this new equipment, the authors demonstrate a variety of Fermi surfaces for various materials, including topological materials and high-T_c superconductors. The equipment is very compact, even smaller than a letter paper size if remove all the cables. Another advantage is that the simple setting allows one to built it at a low cost.

It is interesting to see that the Fermi surface can be accumulated with such a compact setup. The idea itself, converting the emission angle to the location on MCP for the mapping of the Fermi surface, is similar to that of other ARPES types of equipment. The main difference, however, is that FeSuMa is not equipped with the lens for the energy scan and instead uses the retardation field, which eliminates the photoelectrons with energies lower than the Fermi energy, allowing the system to be so simple and low cost.

Only what I worry about most is if or not the contents of this manuscript are suitable for Nature communications, which requires a general interest. Many readers are interested in the Fermi surface for obvious reasons. However, in reality, only a limited number of researchers (mostly the specialists of ARPES) are interested in ARPES equipment. I like the concept of the authors but think that the current manuscript is more fit for a specialized journal such as Review of Scientific Instruments.

Furthermore, I also concern about the following items which are required to become a proper paper, while missing in the current version:

1) The authors emphasize the simplicity of the equipment, and figure 2 and the nicely-made movie for the manuscript indeed look so. However, this information is so limited and the explanation is too brief, as in a proper scientific paper, which is required to give enough info for others to reproduce the same results. For example, how do the three lenses work? Why the 4th is not necessary? If they simply ask one to refer to other ARPES techniques, then it will lose the novelty.

2) I could understand how they map Fermi surfaces, but cold not how they take the energy dispersion with the same technique. The authors should tell the readers more details of the technique associated with the following sentences, perhaps using equations or illustrations: "We record a series of 2D angular distributions, each corresponding to the threshold energies lower than the one used to record FS map, thus scanning the energy interval of interest. The difference between two adjacent measurements would represent the angular distribution corresponding to the particular binding energy."

3) Related to the above 1), the energy dispersions in Fig.4 are largely distorted. Please explain the reason. It might indicate the limitation of the current technique.

4) Some of the Fermi surface maps in Fig. 3 are also severely distorted. What is the reason?

Reviewer #3 (Remarks to the Author):

The manuscript by Borisenko et al. reports a new ARPES apparatus that can map out the equi-energy photoelectron intensity in the k_x and k_y plane with very high efficiency. Usually, high-resolution ARPES spectrometer consists of an electron lens and a hemispherical analyzer; everybody has so far believed that both are essential to achieve high-precision ARPES measurements. However, the authors have shown for the first time that utilization of the electron lens and retarding voltage system (by skipping the use of hemispherical analyzer) is sufficient to map out the whole Fermi surface, breaking a commonly accepted view that the hemispherical analyzer is essential to achieve high-resolution intensity mapping. This result is surprising for the ARPES community, and I like the basic idea of this new measurement system. Also, this system would have a potential to bring about a revolution on the standard ARPES experiments in the future. Although I don't know to what extent the non-ARPES specialists are interested in this work, as an ARPES expert, I would like to recommend the publication of this manuscript in Nature Communications after the authors consider my comments listed below.

1) I think that the electron lens part is essentially similar to those equipped in the commercially available standard ARPES analyzers such as Omicron-Scienta DA30 and MBS-A1. If there exist some intrinsic differences besides the size of a lens system, please specify.

2) Since this apparatus is very simple, I'm not quite convinced that to what extent the calibration of angles vs wave vectors was accurately carried out. For example, the ARPES-derived Dirac-cone band dispersion of Bi_2Te_3 shown in Fig. 4a appears to be highly distorted around the Dirac point and asymmetric with respect to the Gamma point. This behavior is unphysical. I suggest the authors to elaborate on this point.

3) Related to the above question, I would like to know to what extent the ARPES image and dispersion obtained with this new analyzer is similar to or different from the those obtained with the standard ARPES analyzers. Can the authors perform this direct comparison on a specific sample?

4) Minor point: can the authors show raw photoelectron angular distribution as a function of retarding voltage and show more intuitively how the differential ARPES intensity maps displayed in Figs. 4a-d were obtained from the raw data?

Response to referee reports.

Reviewer #1 (Remarks to the Author):

This manuscript presents a simple ARPES equipment mostly focusing on the Fermi surface mapping by photoemission; the authors name it FeSuMa (Fermi Surface Mapper). With this new equipment, the authors demonstrate a variety of Fermi surfaces for various materials, including topological materials and high-T_c superconductors. The equipment is very compact, even smaller than a letter paper size if remove all the cables. Another advantage is that the simple setting allows one to built it at a low cost.

It is interesting to see that the Fermi surface can be accumulated with such a compact setup. The idea itself, converting the emission angle to the location on MCP for the mapping of the Fermi surface, is similar to that of other ARPES types of equipment. The main difference, however, is that FeSuMa is not equipped with the lens for the energy scan and instead uses the retardation field, which eliminates the photoelectrons with energies lower than the Fermi energy, allowing the system to be so simple and low cost.

Only what I worry about most is if or not the contents of this manuscript are suitable for Nature communications, which requires a general interest. Many readers are interested in the Fermi surface for obvious reasons. However, in reality, only a limited number of researchers (mostly the specialists of ARPES) are interested in ARPES equipment. I like the concept of the authors but think that the current manuscript is more fit for a specialized journal such as Review of Scientific Instruments.

We thank the Referee for careful reading of the manuscript and positive words about our instrument. We believe that indeed many general readers are interested in the Fermi surface and they will be happy to learn that from now on it is possible to map it even in 3D and more effectively.

Furthermore, I also concern about the following items which are required to become a proper paper, while missing in the current version:

1) The authors emphasize the simplicity of the equipment, and figure 2 and the nicely-made movie for the manuscript indeed look so. However, this information is so limited and the explanation is too brief, as in a proper scientific paper, which is required to give enough info for others to reproduce the same results. For example, how do the three lenses work? Why the 4th is not necessary? If they simply ask one to refer to other ARPES techniques, then it will lose the novelty.

It may seem astonishing, but the setup is indeed as simple as described in the paper and shown in the animation (Ref. 4). In the meantime, we have also obtained a US patent, and the detailed description is also available online in English (updated Ref. 5). At the end of this description, an example with all dimensions and voltages is given so the results can be reproduced. We use a minimal number of lens elements to keep the design simple. This simplicity contributes to the high resolution of FeSuMa, as the imperfection of any additional element would lead to distortions of the field. On the advice of the Reviewer, we describe the lens in more detail, mentioning the well known example of Einzel lens which consists of exactly three elements, and update the Ref. 5 with the link to the patent description with the detailed example in English.

2) I could understand how they map Fermi surfaces, but could not how they take the energy dispersion with the same technique. The authors should tell the readers more details of the technique associated with the following sentences, perhaps using equations or illustrations: "We record a series of 2D angular distributions, each corresponding to the threshold energies lower

than the one used to record FS map, thus scanning the energy interval of interest. The difference between two adjacent measurements would represent the angular distribution corresponding to the particular binding energy."

Following the recommendation of both Referees, we are including one new figure in the main part and two figures in the supplemental information, along with a corresponding discussion. We hope that the procedure has now been clarified.

3) Related to the above 1), the energy dispersions in Fig.4 are largely distorted. Please explain the reason. It might indicate the limitation of the current technique.

4) Some of the Fermi surface maps in Fig. 3 are also severely distorted. What is the reason?

The intention was to present the raw data in the angular scale of the spectrometer. Every analyzer will yield a „distorted“ map if the normal to the surface will not coincide with the lens axis. In the present version we explain this effect more in the Supplementary Information and show the map in momentum scale in Fig. S1. Spectra from Fig.4 a (now Fig. 5) are shown now in momentum scale and along high symmetry directions.

Reviewer #3 (Remarks to the Author):

The manuscript by Borisenko et al. reports a new ARPES apparatus that can map out the equi-energy photoelectron intensity in the k_x and k_y plane with very high efficiency. Usually, high-resolution ARPES spectrometer consists of an electron lens and a hemispherical analyzer; everybody has so far believed that both are essential to achieve high-precision ARPES measurements. However, the authors have shown for the first time that utilization of the electron lens and retarding voltage system (by skipping the use of hemispherical analyzer) is sufficient to map out the whole Fermi surface, breaking a commonly accepted view that the hemispherical analyzer is essential to achieve high-resolution intensity mapping. This result is surprising for the ARPES community, and I like the basic idea of this new measurement system. Also, this system would have a potential to bring about a revolution on the standard ARPES experiments in the future. Although I don't know to what extent the non-ARPES specialists are interested in this work, as an ARPES expert, I would like to recommend the publication of this manuscript in Nature Communications after the authors consider my comments listed below.

We are grateful to the Reviewer for accurately evaluating our manuscript and recommending it for publication.

1) I think that the electron lens part is essentially similar to those equipped in the commercially available standard ARPES analyzers such as Omicron-Scienta DA30 and MBS-A1. If there exist some intrinsic differences besides the size of a lens system, please specify.

Yes, the lens is similar, but in our case it actually consists of only three elements, as shown in Fig.2.

2) Since this apparatus is very simple, I'm not quite convinced that to what extent the calibration of angles vs wave vectors was accurately carried out. For example, the ARPES-derived Dirac-cone band dispersion of Bi₂Te₃ shown in Fig. 4a appears to be highly distorted around the Dirac point and asymmetric with respect to the Gamma point. This behavior is unphysical. I suggest the authors to elaborate on this point.

In spite of simplicity of our instrument, the angles are accurately calibrated for all modes. We have initially presented the raw data in the angular scale of the spectrometer. Those maps which were

taken at off-normal conditions can be recalculated in momentum scale, as usual. Following the recommendation of the Reviewer, we show one example in Fig. S1 and plot now the cuts from Fig. 4a in momentum scale and in high-symmetry directions.

3) Related to the above question, I would like to know to what extent the ARPES image and dispersion obtained with this new analyzer is similar to or different from the those obtained with the standard ARPES analyzers. Can the authors perform this direct comparison on a specific sample?

All results are very similar to the ones obtained using conventional analyzers. Following the recommendation of the Reviewer we show one example of direct comparison in Fig. S4. With FeSuMa we simply „slice“ the (k,w) -space differently. Horizontally instead of vertically (new Fig. 4). Our unit of information is angle-angle distribution, whereas the one of conventional analyzers is angle-energy distribution. This is reflected in the corresponding resolutions.

4) Minor point: can the authors show raw photoelectron angular distribution as a function of retarding voltage and show more intuitively how the differential ARPES intensity maps displayed in Figs. 4a-d were obtained from the raw data?

We have presented the requested raw data in the new Fig. 4 and in Figs. S2, S3, and appended the corresponding discussion.

REVIEWERS' COMMENTS

Reviewer #1 (Remarks to the Author):

In the previous round of my review, I have requested to add more details of the experimental technique introduced in the manuscript works, especially about the energy scan of ARPES spectra. In the new version of the manuscript, the authors added Fig. 4 and Fig. S3, which show the original 3D and 2D maps and the resulting energy distribution maps obtained by energy derivative. With these new figures, the concept of the equipment demonstrated in the manuscript gets much clearer. The author also compared the data taken by their new equipment with the data measured by the standard hemispherical analyzer, which clarifies how the Fermi surface mapping is much more efficient in the former. The new equipment invented by the authors is very simple, so it provides the possibility that the ARPES technique will get a more common experimental method in condensed matter physics. In that sense, this manuscript might positively impact a wide area of physical society. However, I still feel that the contents are so technical that Nature communications are not a suitable journal for publication, and the more specialized journal is appropriate. My stance has been unchanged since the 1st round of my review, which is also my conclusion on this manuscript.

Reviewer #3 (Remarks to the Author):

I appreciate the authors for their effort to revise the manuscript according to the reviewers' suggestions. I think that the authors have satisfactorily answered to all of my questions regarding the structure of electron lens, calibration of angle vs wave vectors, comparison with data obtained with a regular analyzer, and data processing. The authors have also provided sufficient information related to these points in the revised manuscript. I would recommend the publication of this manuscript in Nature Communications in its present form.

REVIEWERS' COMMENTS

Reviewer #1 (Remarks to the Author):

In the previous round of my review, I have requested to add more details of the experimental technique introduced in the manuscript works, especially about the energy scan of ARPES spectra. In the new version of the manuscript, the authors added Fig. 4 and Fig. S3, which show the original 3D and 2D maps and the resulting energy distribution maps obtained by energy derivative. With these new figures, the concept of the equipment demonstrated in the manuscript gets much clearer. The author also compared the data taken by their new equipment with the data measured by the standard hemispherical analyzer, which clarifies how the Fermi surface mapping is much more efficient in the former. The new equipment invented by the authors is very simple, so it provides the possibility that the ARPES technique will get a more common experimental method in condensed matter physics. In that sense, this manuscript might positively impact a wide area of physical society. However, I still feel that the contents are so technical that Nature communications are not a suitable journal for publication, and the more specialized journal is appropriate. My stance has been unchanged since the 1st round of my review, which is also my conclusion on this manuscript.

We thank the reviewer for their careful review and recognition that we have answered all previous questions and taken into account all comments and suggestions. We have also changed the abstract and introduction to emphasise the overall importance of our findings and to make them less technical.

Reviewer #3 (Remarks to the Author):

I appreciate the authors for their effort to revise the manuscript according to the reviewers' suggestions. I think that the authors have satisfactorily answered to all of my questions regarding the structure of electron lens, calibration of angle vs wave vectors, comparison with data obtained with a regular analyzer, and data processing. The authors have also provided sufficient information related to these points in the revised manuscript. I would recommend the publication of this manuscript in Nature Communications in its present form.

We are pleased with the accurate assessment of our work and have made changes in response to this reviewer's earlier comments and suggestions. We are grateful for the recommendation to publish the work as it stands.